# The role of neighbourhood socioeconomic status in large for gestational age

**Farid Boubred[1], Vanessa Pauly[2,3], Fanny Romain[2], Guillaume Fond[2,3], Laurent Boyer [2,3]***

**1** Neonatal Unit, C2 VN, Hospital University La Conception, APHM, AMU, Marseille, France, **2** Public Health and Medical Information Department, APHM, Marseille, France, **3** EA 3279: CEReSS—Health Service Research and Quality of Life Center, AMU, Marseille, France

* laurent.boyer@ap-hm.fr

## Abstract

### Objective

To determine whether neighbourhood socioeconomic status (SES) was associated with large for gestational age (LGA) while considering key sociodemographic and clinical confounding factors.

### Setting and patient

All singleton infants whose parents were living in the city of Marseilles, France, between 2013 and 2016.

### Method

Population-based study based on new-born hospital birth admission charts from the French National Uniform Hospital Discharge Data Set Database. LGA infants were compared to appropriate-for-gestational-age (AGA) infants. Multiple generalized logistic model analysis was used to examine factors associated with LGA.

### Results

A total of 43,309 singleton infants were included, and 4,747 (11%) were born LGA. LGA infants were more likely to have metabolic and respiratory diseases and to be admitted to the neonatal intensive care unit. Multiparity, advanced maternal age, obesity and diabetes were associated with an increased risk of LGA. Lower neighbourhood SES was associated with LGA (aOR = 1.24, 95% CI: 1.14; 1.36; p<0.0001) independent of age, diabetes, obesity, maternal smoking and multiparity. The strength of this association increased with maternal age, reaching an aOR of 1.50 (95% CI: 1.26; 1.78; p<0.0001) for women > 35 years old.

### Conclusion

Neighbourhood SES could be considered an important factor for clinicians to better identify mothers at risk of having LGA births in addition to well-known risk factors such as maternal diabetes, obesity and age. The intensification of the association between SES and LGA with

**Data Availability Statement:** Data cannot be shared publicly because of this data are issued from French national medicoadministrative database. Data are available from the ATIH Institutional Data Access (contact via https://www.

atih.sante.fr/) for researchers who meet the criteria for access to confidential data. The data underlying the results presented in the study are available from ATIH (https://www.atih.sante.fr/).

**Funding:** The authors received no specific funding for this work.

**Competing interests:** The authors have declared that no competing interests exist.

increasing maternal age suggests that neighbourhood disadvantage may act on LGA cumulatively over time.

## Background

Large for gestational age (LGA) is a growing public health problem in developed countries [1–3], and the prevalence of infants born LGA varies from 5 to 20% in these countries [4]. Recent studies suggest that the number of LGA infants has increased over the last two decades [2, 5]. The number of LGA births is also expected to rise due to the current epidemic context of obesity and diabetes, two major LGA risk factors [6]. Among women with these metabolic complications, the risk of LGA is increased twofold to threefold [7, 8]. LGA increases the risk of numerous adverse perinatal and birth outcomes, including caesarean delivery, postpartum haemorrhage, shoulder dystocia, perinatal asphyxia, neonatal hypoglycaemia, hyperbilirubinemia and neonatal respiratory distress disorders [9–11]. LGA also affects long-term health outcomes by increasing the risk for cancer [12]; metabolic disorders including overweight, obesity and type 2 diabetes in children and adults [13]; and cardiovascular disease [14]. A better way to identify pregnant women who are at risk of delivering a LGA infant is of particular concern for improving neonatal health outcomes, interrupting the intergenerational chain of metabolic diseases and preserving offspring health and well-being in adulthood [15].

Several factors, including maternal overweight/obesity, excessive gestational weight gain and gestational diabetes mellitus, are known to induce LGA [16, 17]. LGA may also be influenced by maternal age: advanced maternal age (*i.e.*, over 35 years) has been suggested to increase the risk of LGA [18]. Births to women of advanced maternal age have increased over the past few decades, reaching rates of approximately 10% to 20% of births in high-income countries [19, 20]. This age group of women has increased rates of various complications, such as preeclampsia, gestational diabetes mellitus, low birth weight and perinatal mortality, during pregnancy [21–23]. A large body of literature has reported the impact of neighbourhood socioeconomic status (SES) on perinatal and birth outcomes, including low birth weight, small for gestational age (SGA) and preterm birth [24–26]. However, the influence of neighbourhood SES on LGA and combined effects with other LGA factors, including maternal age, has received little attention. Neighbourhood SES may affect LGA through the neighbourhood's effects on maternal behaviours, such as the lack of access to healthy food or opportunities for exercise to combat obesity and diabetes and the low access to high-quality maternity care [27]. To our knowledge, only a few studies in the United States and Canada have addressed this issue with conflicting results, e.g., two studies showing an increased risk of LGA and one study showing a decreased risk of LGA for lower neighbourhood SES [28–30]. Further studies are thus needed to confirm the influence of neighbourhood SES on LGA. Explorations of this issue in other countries and health care systems may provide additional information. France may offer an interesting view, as the French Health system offers universal health coverage guaranteeing access to care for pregnant women regardless of cost [31]. The standards of maternity care are considered high, with a total of 7 recommended prenatal examinations. According to the World Health Organization, prenatal examinations provide an important opportunity to prevent and manage concurrent diseases through integrated service delivery [32], and France is one of the European countries with the lowest percentages of late prenatal examinations [33, 34]. Medical costs during the perinatal period are covered at a rate of 100% by the French social security system. Investigating the effect of SES on LGA at the whole

population level with universal maternity coverage would therefore be a significant step in the identification of LGA risk factors.

The objective was to determine whether neighbourhood SES was associated with LGA while considering key sociodemographic and clinical confounding factors in a large population-based study using data from the French National Hospital Database system.

## Methodology

### Study design and data source

This population-based cohort study was based on new-born hospital birth admission charts from the French National Uniform Hospital Discharge Database (*Programme de Médicalisation du Système d'Information*, *PMSI*), which systematically collects administrative and medical information, including obstetrical characteristics. This database is compiled based on diagnosis-related groups, and each diagnosis (both maternal and new-born infant) is coded according to the International Classification of Diseases, Tenth Revision (ICD-10) [35]. Infants were classified using diagnosis-related groups related to births. Only singleton births were included in the analysis. The charts of infants with errors on individual linkage or without a maternal address code were excluded. All singleton infants whose parents were living in the city of Marseilles, France, between 2013 and 2016 were eligible for the study regardless of the maternity ward address. This study was part of a regional project aimed at identifying factors influencing perinatal health outcomes. Marseille is the second-largest city of France, with almost 860,000 inhabitants and 13,000 births per year and is one of the French cities with the highest levels of social inequality [36]. The size of the city was sufficient to use a geographic socioeconomic index based on zip codes. Previous studies in Marseilles reported that social inequalities were well captured by zip codes, *i.e.*, inhabitants shared resembling socioeconomic characteristics in the same zip code [37], which has not been demonstrated in other French cities or in studies associating urban and rural populations.

From the PMSI database, we selected all in-hospital stays corresponding to a neonatal diagnosis-related group (DRG n°15 in France and with an age equal to 0 days) and with address codes localized to Marseilles. We excluded in-hospital stays with linking problems, transfers from hospitals in another geographical area, infants from multiple pregnancies or when the number of infants was not specified, and infants who were small for gestational age (birth weights lower than the 10th percentile). Then, we defined two populations: infants who were LGA (birth weights lower than the 10th percentile) and controls who were appropriate for gestational age (AGA).

### Neighbourhood SES

For each birth, maternal SES was evaluated using the available neighbourhood deprivation index (NDI) developed in France (French Deprivation index, FDep09) based on the residential zip code [38]. The NDI was developed from variables included in the 2009 national census data published by the French National Institute of Statistics and Economic Studies (INSEE, *Institut National de la Statistique et des Etudes Economiques)* and was calculated for each district in Marseille (n = 16). The NDI was used as a proxy for the social environment and was the first component used in a principal component analysis involving four socioeconomic variables: median household income, percentage of the population aged 15 years or older that graduated from high school, percentage of blue-collar workers and the unemployment rate of the population aged between 15 and 64 years. The NDI was categorized according to quartiles, from the least (NDI level 1) to the most deprived area (NDI level 4). This index was adjusted to the study period and validated by variables from the national 2015 Census data (**S1 Table**).

## Maternal and neonatal data collection

Maternal and neonatal characteristics were derived from the infant in-hospital stay. Obstetrical data were included in the new-born hospital birth admission chart. Gestational age, birth weight, admissions to the neonatal intensive care unit (NICU) and data on some neonatal morbidities, including metabolic disorders (ICD-10 codes P700, 701, 711, 709, and 719), jaundice (ICD-10 codes P599, P58, and P590), respiratory diseases (ICD-10 codes P22, P24, and P293), perinatal asphyxia with neonatal encephalopathy (ICD-10 codes P21 and P916) and neonatal death up to day 28, were collected. Preterm birth was defined as gestational age less than 37 weeks. SGA, AGA and LGA were defined as birth weights lower than the 10th percentile, within the 10th and 90th or higher than the 90th percentile, respectively, based on the national French growth chart(19).

Data on the following maternal comorbidities were also searched in the infant in-hospital stay: maternal hypertension or preeclampsia (ICD-P000); gestational diabetes mellitus, including maternal diabetes and gestational diabetes mellitus codes (ICD-P700 and P701); obesity (ICD-10 code E66*: body mass index greater than 30); multiparity; mode of delivery (ICD-10 code P034 for caesarean section); and maternal smoking, through the proxy ICD 10 code P042, which is 'New-born affected by maternal use of tobacco.' Due to coding instructions, parity is only available for vaginal deliveries. Data on maternal age were collected after linkage with maternal hospital admission charts. For infants with unavailable linkage, we replaced the corresponding maternal age with the mean age of the mothers who delivered in the same maternity ward in the same year.

## Statistical analysis

Descriptive data for sociodemographic and clinical characteristics are presented as frequencies and percentages. First, we described and compared the characteristics of the LGA vs AGA infants as well as the characteristics according to the NDI (Table 1). Univariate analyses were performed using a generalized logistic model with the hospital or maternity ward as a random effect to take into account correlations of data due to hospital clustering (Table 2). To determine whether the NDI was an independent factor associated with LGA, we performed stepwise logistic regression analyses. Because data on maternal age was absent for 10% of women, we performed an analysis with imputation to the mean (*i.e.*, replacement of maternal age by the mean age of the mothers who delivered in the same maternity ward in the same year) and an analysis excluding mothers with missing data. Interaction effects were tested between the different variables, particularly the interaction between maternal age and NDI levels presented in Fig 1. Gestational diabetes mellitus and obesity were not included in the same model because of multicollinearity. Because multiparity data were only available for vaginal deliveries, another model for sensitivity analysis was computed only on vaginal delivery data to include multiparity in the analysis. All multivariate analyses are presented in Table 3. Odds ratios (ORs) with 95% confidence intervals (95% CIs) were calculated. Statistical significance was defined as $p<0.05$. The statistical analysis was performed with SAS 9·4 (SAS Institute), and a univariate/multivariate generalized logistic regression model was performed using PROC GLIMMIX in SAS®.

## Ethics

Our institution (Assistance Publique—Hôpitaux de Marseille) was granted access to the French National Uniform Hospital Discharge Data Set Database by the ATIH (Agence Technique d'Information sur l'Hospitalisation https://www.atih.sante.fr/), the organization in charge of this database in France, in compliance with French law after a deliberation of the French Commission for Data Protection and Liberties (CNIL). In addition, research on retrospective data is excluded from the framework of the French Law Number 2012–300 of 5

**Table 1. Maternal and neonatal characteristics of appropriate-for-gestational-age (AGA) and large-for-gestational-age (LGA) infants.**

| Perinatal characteristics | Total population | LGA | AGA | p |
|---|---|---|---|---|
| | (n = 43,309) | (n = 4747) | (n = 38,562) | |
| **Obstetrical data** | | | | |
| **Maternal age, n (%)** | | | | < 0.0001 |
| < 25 y | 10,781 (24.9) | 1054 (22.2) | 9727 (25.2) | |
| 25–30 y | 11,362 (26.2) | 1175 (24.7) | 10,187 (26.4) | |
| 30–35 y | 12,449 (28.7) | 1392 (29.3) | 11,057 (28.7) | |
| ≥ 35 y | 8717 (20.1) | 1126 (23.7) | 7591 (19.7) | |
| **Obesity, n (%)** | 2108 (4.9) | 385 (8.1) | 1723 (4.5) | <0.0001 |
| **Multiparity*, n (%)** | 18,672/31,858 | 2183/2983 | 16,489/28,875 | <0.0001 |
| | (58.6) | (73.2) | (57.1) | |
| **GDM**, n (%)** | 2794 (6.4) | 643 (13.5) | 2151 (5.6) | <0.0001 |
| **Preeclampsia, n (%)** | 777 (1.8) | 99 (2.1) | 678 (1.8) | 0.11 |
| **Maternal smoking, n (%)** | 629 (1.4) | 29 (0.6) | 600 (1.5) | <0.0001 |
| **Caesarean delivery, n (%)** | 8967 (20.7) | 1513 (31.8) | 7454 (19.3) | <0.0001 |
| **Neonatal data** | | | | |
| **GA, mean (SD)** | 39.1 (1.6) | 39.1 (1.7) | 39.1 (1.6) | 0.81 |
| **Preterm birth, n(%)** | 2156 (4.9) | 1909 (4.9) | 207 (5.2) | 0.45 |
| **BW, mean (SD)** | 3374 (467) | 4062 (399) | 3289 (400) | < 0.001 |
| **Sex (female), n (%)** | 20,864 (48.2) | 2340 (49.3) | 18,524 (48) | 0.10 |
| **Congenital malformations, n(%)** | 2705 (6.2) | 330 (6.9) | 2375 (6.1) | 0.07 |
| **NICU admissions, n (%)** | 1993 (6.4) | 304 (6.4) | 1689 (4.4) | <0.001 |
| **Metabolic disorders***, n (%)** | 3021 (7) | 691 (14.5) | 2330 (6) | <0.001 |
| **Jaundice, n (%)** | 3086 (7.1) | 341 (7.2) | 2745 (7.1) | 0.92 |
| **Respiratory disease, n (%)** | 3045 (7) | 450 (9.5) | 2595 (6.7) | <0.001 |
| **Perinatal asphyxia, n (%)** | 1675 (3.9) | 189 (4) | 1489 (3.7) | 0.42 |
| **Neonatal deaths, n (‰)** | 45 (1.03) | 7 (1.47) | 38 (0.98) | 0.32 |
| **Maternal SES** | | | | |
| **NDI, n (%)** | | | | <0.001 |
| **1 least deprived** | 11,845 (27.3) | 1181 (24.9) | 10,664 (27.6) | |
| **2** | 9702 (22.4) | 1045 (22) | 8657 (22.4) | |
| **3** | 9218 (21.3) | 975 (20.5) | 8243 (21.4) | |
| **4 most deprived** | 12,544 (28.9) | 1546 (32.6) | 10,998 (28.5) | |

BW: birth weight (g); GA: gestational age (weeks); GDM: gestational diabetes mellitus; metabolic disorders: hypoglycaemia and hypocalcaemia; NICU: neonatal intensive care unit; NDI: neighbourhood deprivation index.

*Multiparity was calculated for infants delivered vaginally (data not available for caesarean delivery).

**GDM: gestational diabetes mellitus.

***metabolic disorders: hypoglycaemia and hypocalcaemia

March 2012 relating to the research involving human participants, as modified by the Order Number 2016–800 of 16 June 2016. Neither French competent authority (Agence Nationale de Sécurité du Médicament et des Produits de Santé, ANSM) approval nor French ethics committee (Comités de Protection des Personnes, CPP) approval is required in this context.

## Results

The study period included 47,331 singleton infants after exclusion of 4,341 infants' charts due to errors or missing data, the absence of the mother's address code, and multiple pregnancies.

**Table 2. Perinatal characteristics of the population according to the neighbourhood deprivation index.**

| Perinatal characteristics | Neighbourhood deprivation index | | | | p |
|---|---|---|---|---|---|
| | Q1 | Q2 | Q3 | Q4 | |
| | Least deprived | | | Most deprived | |
| | (n = 11,845) | (n = 9702) | (n = 9218) | (n = 12,544) | |
| **Maternal data** | | | | | |
| **Maternal age, n (%)** | | | | | <0.001 |
| < 25 y | 2174 (18.3) | 2222 (22.9) | 2653 (28.8) | 3732 (29.7) | |
| 25–30 y | 2892 (24.4) | 2720 (28) | 2293 (24.9) | 3457 (27.5) | |
| 30–35 y | 4036 (34.1) | 2867 (29.5) | 2491 (27) | 3055 (24.3) | |
| ≥ 35 y | 2743 (23.1) | 1893 (19.5) | 1781 (19.3) | 2300 (18.3) | |
| **Obesity, n (%)** | 205 (1.7) | 444 (4.6) | 570 (6.2) | 889 (7.1) | <0.001 |
| **Multiparity\*, n (%)** | 4857/9147 | 4034/7241 | 3877/6527 (59.4) | 5904/8943 (66) | <0.001 |
| | (53.1) | (55.1) | | | |
| **Gestational diabetes mellitus, n (%)** | 475 (4) | 573 (5.9) | 633 (6.9) | 1113 (8.9) | <0.001 |
| **Preeclampsia, n (%)** | 170 (1.4) | 192 (1.9) | 168 (1.8) | 247 (1.9) | 0.005 |
| **Maternal smoking, n (%)** | 92 (0.8) | 188 (1.9) | 175 (1.9) | 174 (1.4) | <0.001 |
| **Caesarean delivery, n (%)** | 2146 (18.1) | 1957 (20.2) | 1893 (20.5) | 2639 (21) | <0.001 |
| **Neonatal data** | | | | | |
| **GA, mean (SD)** | 39.20 (1.5) | 39.13 (1.6) | 39.16 (1.6) | 39.08 (1.7) | <0.001 |
| **BW, mean (SD)** | 3372 (449) | 3365 (467) | 3378 (458) | 3379 (489) | 0.33 |
| **Preterm birth, n (%)** | 546 (4.6) | 501 (5.1) | 425 (4.6) | 684 (5.4) | < 0.01 |
| **Sex (female), n (%)** | 5666 (47.8) | 4672 (48.1) | 4453 (48.3) | 6073 (48.4) | 0.82 |
| **LGA, n (%)** | 1181 (9.9) | 1045 (10.8) | 975 (10.6) | 1546 (12.3) | <0.001 |
| **LGA with diabetes mellitus/GDM, n (%)** | 110 (0.9) | 136 (1.4) | 137 (1.5) | 260 (2.1) | <0.001 |
| **Congenital malformations, n (%)** | 755 (6.4) | 655 (6.7) | 600 (6.5) | 695 (5.5) | 0.001 |
| **NICU admissions, n (%)** | 433 (3.7) | 509 (5.2) | 425 (4.6) | 626 (5) | <0.001 |
| **Neonatal death, n (‰)** | 9 (0.76) | 9 (0.97) | 9 (0.97) | 18 (1.43) | 0.3 |

BW: birth weight (g); GA: gestational age (weeks); GDM: gestational diabetes mellitus; NICU: neonatal intensive care unit; Neonatal death: death during the first 28 days of life.

\*Multiparity: calculated for infants delivered vaginally.

After the exclusion of 4,022 infants for SGA, 43,309 were included in this study, of which 4,747 (10.9%) were LGA and 38,562 were AGA (Fig 2). The linkage between mother and infants was available for 90% of the infants. The obstetrical and neonatal characteristics of the study population are detailed in **Table 1**. The rates of infants born to mothers of advanced age (*i.e.*, over 35 years), gestational diabetes mellitus and preeclampsia were 20.1%, 6.4% and 1.8%, respectively. Approximately 5% of the infants in the study population were born preterm (gestational age less than 37 weeks).

## LGA population characteristics

The comparison of AGA and LGA infants is presented in **Table 1**. In comparison with AGA infants, LGA infants were more likely to be born to older (maternal age above 35 years) and multiparous mothers (p<0.0001). The rate of gestational diabetes mellitus was 2.5-fold higher (p<0.0001) among LGA infants than among AGA infants. In contrast, the rate of infants born to tobacco-smoking mothers was 3-fold lower (p<0.0001) in the LGA group. While the mean GA and preterm birth rate were comparable between groups, NICU admissions were

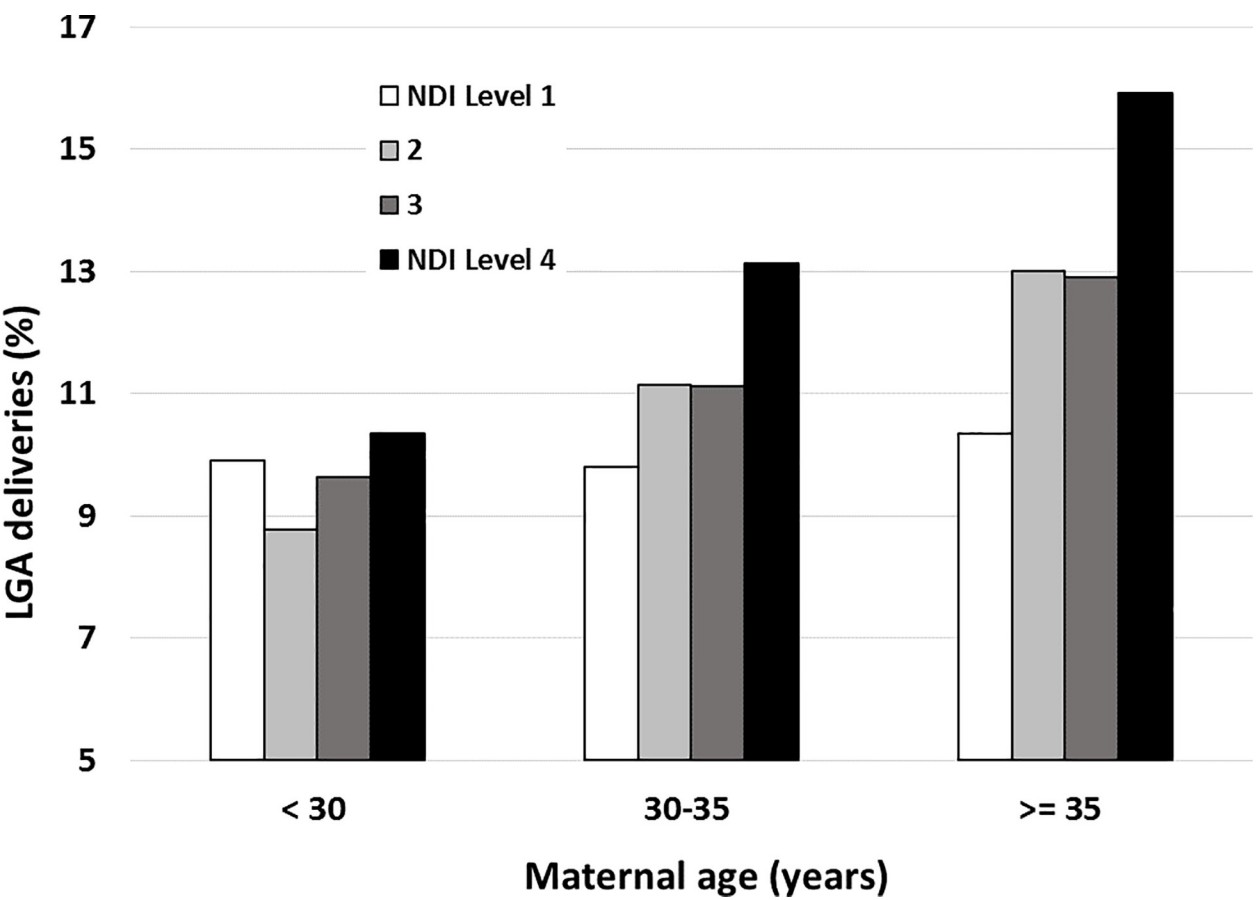

**Fig 1. LGA, neighbourhood deprivation index (NDI) and maternal age.** The rate of infants born LGA increased gradually with advancing maternal age, and the NDI reached the highest level among infants born to mothers aged 35 years and living in the most deprived neighbourhoods.

significantly higher for LGA infants than for AGA infants (p<0.0001), with higher rates of hypoglycaemia, hypocalcaemia and respiratory disorders (p< 0.0001).

## Neighbourhood deprivation and perinatal outcomes

Perinatal outcomes according to the NDI are detailed in **Table 2**. Approximately 30% of the population lived in the most deprived neighbourhoods (NDI level 4). The proportion of LGA varied by NDI level (12.3% in NDI level 4 vs 9.9% in NDI level 1, p <0.0001). Infants born to mothers living in the most deprived areas were more likely to be born preterm (p<0.001). Infants from NDI levels 2, 3 and 4 were more frequently exposed to maternal smoking than those from NDI level 1 (p < 0.001). In contrast, in less deprived neighbourhoods, infants were more likely to be born to older (p<0.0001) and nulliparous (p<0.0001) mothers. The mean GA decreased with NDI, and the mean birth weight was unchanged due to a higher rate of LGA. Similarly, gestational diabetes mellitus (p<0.0001) and LGA associated with gestational diabetes mellitus (p<0.01) were 2-fold higher in NDI level 4 than in NDI level 1.

## Factors associated with LGA

Multivariate analyses are detailed in **Table 3**. NDI, advanced maternal age (maternal age ≥ 35 years), gestational diabetes mellitus and obesity were significantly associated with LGA

**Table 3. Factors associated with LGA delivery.** Odds ratios (95% CI) from the stepwise logistic regression models.

| | Model with imputation of maternal age* N = 43,309 | | Model without imputation of maternal age N = 38,547 | | Model with obesity instead of GDM N = 38,547 | | Model including multiparity only for vaginally delivered infants N = 31,858 | |
|---|---|---|---|---|---|---|---|---|
| | aOR (95% CI) | *P* | aOR (95% CI) | *P* | aOR (95% CI) | *P* | aOR (95% CI) | *P* |
| **Maternal age** | | <0.001 | | <0.001 | | <0.001 | | <0.01 |
| < 25 y | 1 [Reference] | | 1 [Reference] | | 1 [Reference] | | 1 [Reference] | |
| 25–30 y | 1.06 (0.97 - 1.16) | 0.17 | 1.40 (1.25 - 1.57) | <0.001 | 1.43 (1.27 - 1.60) | <0.001 | 1.26 (1.10 – 1.44) | <0.001 |
| 30–35 y | 1.15 (1.05 - 1.26) | 0.002 | 1.52 (1.36 -1.70) | <0.001 | 1.59 (1.42 -1.77) | <0.001 | 1.20 (1.05 – 1.37) | <0.01 |
| > 35 y | 1.28 (1.17 - 1.41) | <0.001 | 1.70 (1.51 -1.91) | <0.001 | 1.83 (1.63 -2.05) | <0.001 | 1.24 (1.07 – 1.42) | <0.01 |
| **Gestational diabetes mellitus** | 2.61 (2.36 - 2.88) | <0.001 | 2.49 (2.24 - 2.76) | <0.001 | - | - | 1.92 (1.67 - 2.20) | <0.001 |
| **Obesity** | - | - | - | - | 1.87 (1.66 - 2.12) | <0.001 | - | - |
| **Multiparity** | - | - | - | - | - | - | 1.95 (1.79 - 2.14) | <0.001 |
| **Maternal smoking** | 0.38 (0.26 - 0.55) | <0.001 | 0.37 (0.25 - 0.55) | <0.001 | 0.37 (0.25 - 0.54) | <0.001 | 0.38 (0.23 - 0.61) | <0.001 |
| **NDI** | | <0.001 | | <0.001 | | <0.001 | | 0.047 |
| **1 least deprived** | 1 [Reference] | | 1 [Reference] | | 1 [Reference] | | 1 [Reference] | |
| **2** | 1.08 (0.99 - 1.18) | 0.08 | 1.12 (1.02 - 1.23) | 0.03 | 1.13 (1.02 - 1.24) | 0.015 | 1.12 (1.00 - 1.18) | 0.04 |
| **3** | 1.08 (0.96 - 1.16) | 0.24 | 1.11 (1.00 - 1.23) | 0.05 | 1.12 (1.01 - 1.23) | 0.036 | 1.05 (0.93 - 1.18) | 0.39 |
| **4 most deprived** | 1.24 (1.14 - 1.36) | <0.001 | 1.28 (1.16 - 1.41) | <0.001 | 1.28 (1.16 - 1.41) | <0.001 | 1.15 (1.03 - 1.29) | 0.01 |
| **Goodness of fit (AIC)** | 29.606 | | 25.864 | | 26.040 | | 19.364 | |

NDI: neighbourhood deprivation index; aOR: adjusted OR.

*A significant interaction was found between maternal age and NDI: The strength of the association between SES and LGA increased with maternal age, reaching an aOR of 1.50 (95% CI: 1.26; 1.78; p<0.0001) for women older than 35 years in ND4.

(p<0.001). Infants born to mothers living in deprived neighbourhoods (NDI level 4) had 25% higher odds of being LGA (aOR = 1.24, 95% CI: 1.14; 1.36; p<0.0001) after adjustment for maternal age, gestational diabetes mellitus, obesity, maternal smoking and multiparity. The strength of the association between SES and LGA increased with maternal age, reaching an aOR of 1.50 (95% CI: 1.26; 1.78; p<0.0001) for women older than 35 years in NDI level 4 (significant interaction). Fig 1 illustrates how LGA varies according to maternal age and the NDI. The differences in the rate of LGA between NDI levels 1 and 4 increased with advanced maternal age, with differences ranging from 4% to 54% in mothers under the age of 30 years relative to those aged 35 years and older (p < 0.001). SES was still significant after the inclusion of multiparity in the sensitivity analysis performed only on vaginal deliveries (n = 31,858), with an aOR = 1.15 (95% CI: 1.03–1.29, p < 0.001); multiparity was associated with LGA with an aOR = 1.19 (95% CI: 1.79–2.14). Maternal smoking was associated with lower odds of LGA (p<0.001).

## Discussion

### Main findings

The present findings can be summarized as follows. Between 2013 and 2016, 47,331 singleton infants whose parents were living in the city of Marseille were identified in the French National Hospital Database system. The prevalence of LGA varied according to neighbourhood SES, reaching 12% in the most deprived neighbourhoods. Lower neighbourhood SES was associated with LGA independent of maternal age, gestational diabetes mellitus, obesity, maternal smoking and multiparity. The strength of this association increased with maternal age, reaching an aOR of 1.50 for women older than 35 years of age.

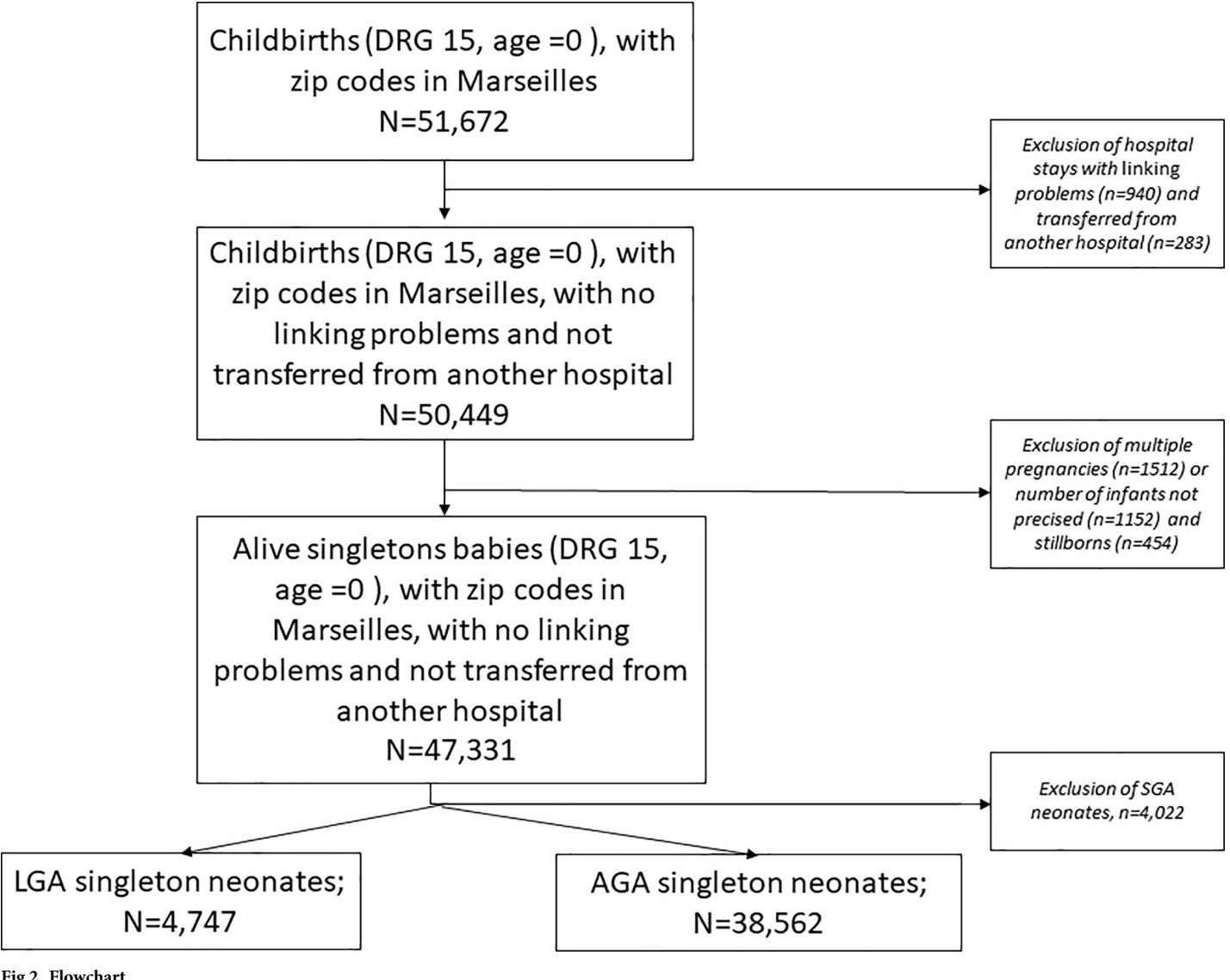

**Fig 2. Flowchart.**

## Interpretation

The first important finding is the significant association between neighbourhood SES and LGA in a large population-based study, complementing the previous results on the effect of SES on other perinatal outcomes such as low birth weight, SGA and preterm birth [16, 39, 40]. To date, the association between SES and LGA has rarely been investigated, especially in industrialized countries with developed health care systems. Our findings confirm the results of two previous studies [28,30] in another socioeconomic and cultural contexts and with a large population but contradict the results of Shankardass et al [29], which did not adjust their findings to potential confounding characteristics, such as in our study. Importantly, this association remained significant after adjustment for metabolic disturbances, suggesting that neighbourhood SES could be considered relevant information for clinicians to better identify mothers at risk of having LGA births in addition to well-known risk factors such as gestational diabetes mellitus, maternal obesity and advanced maternal age. The association between

neighbourhood SES and LGA may be explained by mechanisms such as poor health behaviours, endocrine disruptors and race/ethnicity characteristics. Future studies should explore the association between neighbourhood SES and inadequate diets (*e.g.*, micronutrient and vitamin deficiencies [41]), endocrine disruptor exposure and ethnic characteristics. For ethical concerns, it is not possible to record ethnicity in the French databases. However, some previous studies have suggested that gypsy, Romanian and sub-Saharan migrant children may be at higher risk of LGA [42].

The LGA rate in deprived neighbourhoods was 25% higher than that in less deprived neighbourhoods. This finding confirms that LGA remains an important health disparity and that universal health coverage is not sufficient to counter the multiple factors that contribute to health and health care disparities [43], namely, the levels and distribution of income, education, housing, nutrition and safety [43, 44, 45]. In accordance with this finding, a previous French study performed in Paris reported insufficient healthcare follow-up among women with low SES [46]. Future studies should better explore economic, social, or environmental disadvantages, and targeted interventions should be proposed to reduce health disparities in these deprived neighbourhoods [47].

Finally, we found a significant interaction between maternal age and neighbourhood SES on the risk of LGA. Advanced maternal age is known to adversely affect birth and neonatal outcomes, but its effects on LGA remain poorly explored [21–23]. Kenny LC *et al.* reported an increase in LGA births among older women (> 30 years), and as we observed in our study, they found larger effects among women living in deprived neighbourhoods [18]. The underlying mechanism is unclear. Although we do not have information on the duration of residence, it may be reasonably hypothesized that LGA risk may increase with the duration of exposure to the abovementioned SES-associated factor. This hypothesis could be supported by previous studies on the life-course accumulation of neighbourhood disadvantage and its impact on health [48]. It can be speculated that older mothers with low SES have a cumulative disease risk over time, including repeated adverse health events that are stressful, diabetes, obesity and multiparity, which in turn promote LGA [3]. An association has been reported between older mothers and low SES and several risky behavioural factors, such as psychosocial issues, smoking practices, alcohol exposure, and stress, which should be specifically targeted early in pregnancy [49–51]. If this cumulative risk hypothesis were to be confirmed, our results would suggest that preventive efforts in current and individual conditions are not sufficient and that early health and social support (individual and/or collective) interventions are also necessary to improve perinatal and birth outcomes.

## Strengths and limitations

The French National Uniform Hospital Discharge Data Set Database does not include some relevant individual factors, including educational status, ethnicity, maternal BMI before or in early pregnancy, maternal body weight gain, hyperglycaemic oral test, maternal nutritional status and data regarding foetal growth. Thus, we were unable to identify how these factors and their severity may mediate the effects of neighbourhood deprivation on LGA. Some comorbidities and clinical characteristics derived from ICD codes could be underreported, such as maternal smoking or obesity. In addition, we did not analyse congenital defects that could affect pregnancy and infant size at birth. Specific studies should explore this issue. Such limitations may nevertheless be mitigated by the fact that our study is the first large population-based study performed in industrialized countries. Our data were extracted from the French national database, which accurately registers hospitalized new-borns, maternal health data and the domicile postal code of the mother's address. All consecutive deliveries in private or public

hospitals providing different levels of health care were collected in this registered database system, which strengthens the findings of the study. Last, the accuracy of geographical methods based on the residential postal code can nevertheless be put into question in studies including urban and rural populations with nonhomogeneous social neighbourhood characteristics. This study avoided such a bias by limiting investigations to a single large city. However, conducting a study in a single French city may limit the generalizability of our findings. Future studies should thus confirm our findings in other geographical areas.

## Conclusions

This study showed that infants born to mothers living in deprived socioeconomic areas had a higher risk of being LGA than those who were born to mothers living in less socioeconomically deprived areas. Neighbourhood disadvantages should be taken into account in routine follow-up care of pregnant women as well as other risk factors to prevent LGA deliveries. The intensification of the association between SES and LGA with increasing maternal age also suggests that neighbourhood disadvantage may act cumulatively on LGA. This finding argues for the integration of early health and social support interventions in the most deprived neighbourhoods to reduce LGA.

## Supporting information

**S1 Table. Comparison of the neighbourhood deprivation index used in the study with related variables from the 2015 French National Census data.**
(DOCX)

## Author Contributions

**Conceptualization:** Farid Boubred, Laurent Boyer.

**Formal analysis:** Vanessa Pauly.

**Methodology:** Farid Boubred, Vanessa Pauly, Fanny Romain, Guillaume Fond, Laurent Boyer.

**Supervision:** Farid Boubred, Fanny Romain.

**Validation:** Farid Boubred, Guillaume Fond.

**Writing – original draft:** Farid Boubred, Laurent Boyer.

**Writing – review & editing:** Vanessa Pauly, Fanny Romain, Guillaume Fond.

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
