## [Decision Letter · Decision Letter 0]

25 Feb 2020

PONE-D-19-26817

The role of neighborhood socioeconomic status in large for gestational age: a nationwide study

PLOS ONE

Dear Dr Boyer,

Thank you for submitting your manuscript to PLOS ONE. After careful consideration, we feel that it has merit but does not fully meet PLOS ONE’s publication criteria as it currently stands. Therefore, we invite you to submit a revised version of the manuscript that addresses the points raised during the review process.

We would appreciate receiving your revised manuscript by 23rd March 2020. To enhance the reproducibility of your results, we recommend that if applicable you deposit your laboratory protocols in protocols.io, where a protocol can be assigned its own identifier (DOI) such that it can be cited independently in the future. For instructions see: http://journals.plos.org/plosone/s/submission-guidelines#loc-laboratory-protocols

We look forward to receiving your revised manuscript.

Kind regards,

Diane Farrar

Academic Editor

PLOS ONE

Journal Requirements:

2. Please provide additional details regarding participant consent. In the ethics statement in the Methods and online submission information, please ensure that you have specified (1) whether consent was informed and (2) what type you obtained (for instance, written or verbal). If your study included minors, state whether you obtained consent from parents or guardians. If the need for consent was waived by the ethics committee, please include this information.

Reviewers' comments:

Reviewer's Responses to Questions

**Comments to the Author**

1. Is the manuscript technically sound, and do the data support the conclusions?

Reviewer #1: Partly

Reviewer #2: Yes

2. Has the statistical analysis been performed appropriately and rigorously? 

Reviewer #1: Yes

Reviewer #2: Yes

3. Have the authors made all data underlying the findings in their manuscript fully available?

Reviewer #1: Yes

Reviewer #2: No

4. Is the manuscript presented in an intelligible fashion and written in standard English?

Reviewer #1: Yes

Reviewer #2: Yes

5. Review Comments to the Author

Reviewer #1: 1. Overall evaluation

This study evaluates the correlation between zip-code level neighborhood socioeconomic status (SES) and the risk of large for gestational age (LGA), focusing on newborn hospital birth admission claims in an urban area – Marseille – extracted from nationally representative administrative data called the French National Uniform Hospital Discharge Database. Statistical strategy is straightforward, applying multivariate logistic regression model. The result shows that lower zip-code level of neighborhood SES tend to be associated with LGA significantly, after adjusting for mother’s age, diabetes, obesity, smoking behavior and multiparity. Further, supposed that mother’s age is a proxy of duration of residency, the author(s) discuss that older mothers living in the area with lower SES might experience a cumulative disease risk over time and therefore, the intensification of the correlation between neighborhood SES and LGA increases along with mother’s age.

I fully agreed with the contributions and novelties highlighted by the author(s), such that this study sheds a light on the issue under the universal maternity coverage in a developed country – France – other than Northern American countries; and also it could provide clinicians additional information to help them understand mothers’ risks of LGA births, along with conventional ones, such as maternal diabetes, obesity and age. However, I think that the author(s) had better to explain carefully about the situation in France as a background to persuade international readers. I would like to provide both major and minor comments as follows. Hopefully, they are helpful for the author(s) to improve this paper.

2. Major Comments

2-1. Title

The current title “The role of neighborhood socioeconomic status in large for gestational age: a nationwide study” is unclear about target population to be analyzed. Further, although this study focused only on an urban area in an urban area – Marseille –, the subtitle (“a nationwide study”) might mislead the readers. So, I strongly suggest the author(s) to revised the title to the one which reflects the contents of this study more precisely.

2-2. More detailed explanation may be helpful for international readers to understand the situation in France as a background information

In the section of “BACKGROUND”, the author(s) had better to provide more detailed explanation how serious LGA births has been becoming, not only in France but also in other developed countries, showing some basic statistics (time-trend LGA births, if possible).

Also, the author(s) emphasizes high standards of maternity care in France due to guaranteed access to care regardless of costs for pregnant women in 14th line of page 3, without any reference. So, the author(s) need to show some reference for this. Then, the author(s) had better to show basic statistics and/or to conduct institutional explanation for proving how high the standards of maternity care in France are, compared to other developed countries like OECD countries.

Adding such more detailed clarifications may help to persuade international readers how important this study is.

2-3. Why Marseille?

As long as I noticed, in the section of “METHODOLOGY”, there are no explanations why the author(s) chose Marseille from a nationwide population-based database in France. Then, from the 2nd bottom line in page 9 through page 10, the author(s) briefly mentioned why they chose an urban large city for this study, such that “Lastly, the accuracy of geographical methods based on residential postal code can nevertheless be put into question in studies including urban and rural populations with nonhomogeneous social neighborhood characteristics. This study avoided such a bias by limiting investigations to a single large city”. The author(s) definitely should reallocate this sentence to the “METHODOLOGY” section. Further, the author(s) should clarify the reason why they chose Marseille among multiple urban areas/cities in France and they have to discuss the possible sampling bias issues on the estimation caused by focusing on a large city like Marseille in “DISCUSSION” section.

2-4. Statistical analysis & results

2-4-1. A flowchart for the procedure of extracting the data to be analyzed is necessary

In the first paragraph of “RESULTS” section in page 6, the author(s) describe how to extract the data to be analyzed in this study. I suggest to bring this explanation to “METHODOLOGY” section and show it by a figure of flowchart. It may be very helpful for the readers to understand how the data is constructed.

2-4-2. Justification for the threshold “p-values<0.2”

In the 7-9th line of page 5, the author(s) said “Variables relevant to the model were selected based on a threshold p<0.2”. Please provide the rationale for choosing this threshold. Otherwise, the selection of explanatory variables seems to be with some ‘intention’. If the author(s) do not have any clear justifications for this, they should to some stepwise logistic regression analyses.

2-4-3. Sensitivity analysis excluding/imputing missing data on maternal age

From the bottom line of page 5 through page 6, the author(s) mentioned the above sensitivity analyses. However, I do not see any results for this. Please show the results of sensitivity analysis, at least in supplementary tables. Also, if the author(s) did some imputation for missing maternal age, they should briefly explain which imputation method they used, either in the main body of text or in the appendix.

2-4-4. Test statistics for regression analyses are necessary to be shown in Table 3

Test statistics for validity of the model should be shown, such as log likelihood, pseudo-R2 values, and Akaike's Information Criterion (AIC).

2-4-5. Univariate results are not necessary to be shown in Table 3

In Table 3, the author(s) show both results of univariate and multivariate logistic regression analyses. However, I do not see any meaning to show univariate results (besides, the results do not seem to be different much). Rather, it is more interesting to show the results of supplementary tables 2 and 3. Besides, in page 7, the author(s) explain the results (ORs) of these supplementary tables in detail. For example, the author(s) said “The strength of the association between SES and LGA increased with maternal age, reaching an aOR of 1.59 (95% CI: 1.26; 1.78 p<0.0001)...”, which cannot be seen in any tables in the main body of manuscript.

2-4-6. How to estimate the results for Figure 1 & supplementary table 3

I do not see how to extract the results of Figure 1 (supplementary table 3). I guess the author(s) introduce interactive terms of maternal age and NDI levels into logistic regression analysis. If possible, please clarify the estimation model in “METHODOLOGY” section. Otherwise, the reader must be lost in a series of explanation about the results of Figure 1 in page 7.

3. Minor Comments

(1) In page 4, the translation of the name of dataset sounds a little strange. How about “the French National Uniform Hospital Discharge Data” or “the French National Uniform Hospital Discharge Database”. I do not think that the author(s) need to repeat “Data Set Database”.

(2) In the 8th line of page 5, what is (Audipog) (19)? Please clarify.

(3) In “RESULTS” in page 6, the author(s) said “The rates of infants born to mothers with advanced age (more than 35 years)” is “51%”, which is different from the value (“20.1% for age 35 and older”) shown in Table 1.

(4) Same as (1), In “RESULTS-LGA population characteristics”, the author(s) said “In comparison with AGA infants, LGA infants were more likely to be born to older (maternal age more than 35 years) and multiparous mothers (p<0.0001)”. However, Table 1 shows an opposite result, such that the ratio of Cesarean delivery tends to be significantly larger for LGA infants so that LGA infants were less likely to be born to multiparous mothers.

(5) At the second line from the bottom of page 6, the author(s) explains about GA (p=0.81). Then, again, the author mentioned about GA at the first line of page 7, such as “While the mean GA and preterm birth rate were comparable”. Better to erase either one.

(6) At the forth line in page 7, the author(s) mentioned “with higher rates of hypoglycemia, hypocalcemia”. Yet, Table 1 does not show the rates of these variables. Please add these in Table 1, if the author(s) use these variables for the further statistical analyses.

(7) In “RESULTS-Neighborhood deprivation and perinatal outcomes” in page 7, the author(s) mentioned “… and to be prenatally exposed to maternal smoking”, which is different from the results shown in Table 2. Table 2 shows that infants born to mothers living in Q2 and Q3 of NDI are more likely to expose to maternal smoking (1.9%), than Q4 (1.4%) and Q1 (0.8%). Please revise the sentence.

(8) Again, in the same section as above (5), the author(s) repeatedly explain about the ratio of LGA. In the 2-3 lines of this section, the author(s) said “The proportion of LGA varies by NDI levels, reaching 12% in NDI level 4”. Then, in the 3-4th lines from the bottom of this section, the author(s) repeatedly explain “…, while we observe a higher rate of LGA with the NDI (12.3% in NDI level 4 vs 9.9% in NDI level 1, p<0.0001)”. Please drop the first part.

(9) As mentioned in my major comments, I do not see any reasons why the author(s) show “Univariate analysis” in Table 3, which is not necessary. Instead, the author(s) should show the results of supplemental table 2 & supplemental data table 3.

Reviewer #2: This manuscript examined associations between neighbourhood-level socioeconomic status and large-for-gestational age (LGA) risk in a sample of 43,309 births between 2013 and 2016 in Marseilles, France. Independent of other variables, lower neighbourhood socioeconomic status was associated with increased risk for LGA status, and this trend was pronounced for older women. This is a well written and interesting article. Specific comments are below.

First, exploring interactions between neighbourhood SES and other variables, and specifically with age, were not developed or justified in the Introduction. Theoretical background justifying the examination of this interaction, and preparing the reader for presentation of interaction findings in the Results, is needed.

Related, the authors note in the Results that “as expected, maternal smoking was a protective factor.” For the reader, this finding is not expected at all because the authors provide no literature discussing the effect of smoking on LGA risk. And smoking during pregnancy is not usually considered a good or “protective” factor, which is what this sentence implies. Please consider revising and providing more context for this finding.

Second, did the authors consider excluding children with congenital defects, which could have affected pregnancy and infant size at birth?

Third, a key and interesting finding is that the effect of neighbourhood socioeconomic status on LGA status is larger for older women compared to younger. The authors also report, however, that older women in the more- and less-deprived neighbourhoods seem to differ in significant ways. Could the authors explicitly characterize the older women in more-deprived neighbourhoods, provide context for those differences, and tie that into clinical implications for these findings?

Fourth, the authors say in the Discussion that “neighbourhood SES may thus be a good proxy for capturing other mechanisms associated with LGA.” The proxies listed are very heterogenous, and as the authors note not well captured in this data set. I would advise caution in making such a statement about using neighbourhood socioeconomic status as a “proxy” for everything from poor health behaviours, endocrine disruptors and race/ethnicity. Please revise.

6. PLOS authors have the option to publish the peer review history of their article (what does this mean?). If published, this will include your full peer review and any attached files.

Reviewer #1: No

Reviewer #2: No

---

## [Decision Letter · Decision Letter 1]

14 Apr 2020

PONE-D-19-26817R1

The role of neighborhood socioeconomic status in large for gestational age

PLOS ONE

Dear Prof Boyer,

Thank you for submitting your manuscript to PLOS ONE. After careful consideration, we feel that it has merit but does not fully meet PLOS ONE’s publication criteria as it currently stands. Therefore, we invite you to submit a revised version of the manuscript that addresses the points raised during the review process.

Firstly you say in the Discussion that “neighbourhood SES may thus be a good proxy for capturing other mechanisms associated with LGA.” The proxies listed are very heterogenous, and as noted not well captured in this data set. I would advise caution in making such a statement about using neighbourhood socioeconomic status as a “proxy” for everything from poor health behaviours, endocrine disruptors and race/ethnicity. Please revise accordingly.”

Secondly I am concerned about the following paragraph: “In line with the previous point, LGA prevalence in deprived neighborhoods was not expected to be so high (i.e., 25% higher than in less deprived neighborhoods) due to the universal health coverage that is supposed to reduce health care inequalities[27]. One explanation is that access to care is a multidimensional concept consisting of both financial and nonfinancial dimensions [43]. Our findings may suggest that nonfinancial barriers may substantially limit access to care despite universal health coverage in this study. In accordance with this hypothesis, a previous French study performed in Paris reported insufficient healthcare follow-up in women with low socioeconomic status[44]. Future studies should better explore nonfinancial barriers to care, and targeted interventions should be proposed for reducing nonfinancial barriers in these deprived neighborhoods.” This paragraph makes it seem as if you are unfamiliar with the vast literature on the social determinants of health and the literature on health care disparities, which identify myriad factors, in addition to health coverage, that contribute to health and health care disparities. Also, LGA prevalence is a health disparity, not a health care disparity, so please consider the contributors to both and revise.

I have a few minor suggestions:

1.            You state “To our knowledge, only a few studies in the United States and Canada have addressed this issue with conflicting results [28-30].” It would be helpful to know what those conflicting results were, and if possible, why the authors believe these studies have produced conflicting results and, in the discussion, how this study addresses these conflicting results.

2.            You should spell out abbreviations the first time they are used (e.g., GDM, AGA). They should consider using fewer abbreviations to improve readability, especially since this is geared to a multi-disciplinary audience who are not likely to be familiar with these abbreviations.

3.            I recommend another round of editing, as there are some grammatical errors.”

We would appreciate receiving your revised manuscript by the 5th May 2020. To enhance the reproducibility of your results, we recommend that if applicable you deposit your laboratory protocols in protocols.io, where a protocol can be assigned its own identifier (DOI) such that it can be cited independently in the future. For instructions see: http://journals.plos.org/plosone/s/submission-guidelines#loc-laboratory-protocols

We look forward to receiving your revised manuscript.

Kind regards,

Diane Farrar

Academic Editor

PLOS ONE

Reviewers' comments:

Reviewer's Responses to Questions

**Comments to the Author**

1. If the authors have adequately addressed your comments raised in a previous round of review and you feel that this manuscript is now acceptable for publication, you may indicate that here to bypass the “Comments to the Author” section, enter your conflict of interest statement in the “Confidential to Editor” section, and submit your "Accept" recommendation.

Reviewer #1: All comments have been addressed

Reviewer #2: All comments have been addressed

2. Is the manuscript technically sound, and do the data support the conclusions?

Reviewer #1: Yes

Reviewer #2: Yes

3. Has the statistical analysis been performed appropriately and rigorously? 

Reviewer #1: (No Response)

Reviewer #2: Yes

4. Have the authors made all data underlying the findings in their manuscript fully available?

Reviewer #1: Yes

Reviewer #2: Yes

5. Is the manuscript presented in an intelligible fashion and written in standard English?

Reviewer #1: Yes

Reviewer #2: Yes

6. Review Comments to the Author

Reviewer #1: I appreciate the author(s) to accept all my suggestions. I do not have any further comments on this article.

Reviewer #2: Thank you for the opportunity to review this manuscript again. The authors have addressed all my comments.

7. PLOS authors have the option to publish the peer review history of their article (what does this mean?). If published, this will include your full peer review and any attached files.

Reviewer #1: No

Reviewer #2: No

---

## [Editor Report · Decision Letter 2]

6 May 2020

The role of neighborhood socioeconomic status in large for gestational age

PONE-D-19-26817R2

Dear Dr. Boyer,

We are pleased to inform you that your manuscript has been judged scientifically suitable for publication and will be formally accepted for publication once it complies with all outstanding technical requirements.

With kind regards,

Diane Farrar

Academic Editor

PLOS ONE

---

## [Editor Report · Acceptance letter]

18 May 2020

PONE-D-19-26817R2 

The role of neighbourhood socioeconomic status in large for gestational age 

Dear Dr. Boyer:

I am pleased to inform you that your manuscript has been deemed suitable for publication in PLOS ONE. Congratulations! Your manuscript is now with our production department. 

With kind regards,

on behalf of

Dr. Diane Farrar 

Academic Editor

PLOS ONE